# Orientation of non-spherical protonated water clusters revealed by infrared absorption dichroism

Jan O. Daldrop[1], Mattia Saita[1], Matthias Heyden[2], Victor A. Lorenz-Fonfria[3], Joachim Heberle[1] & Roland R. Netz[1]

Infrared continuum bands that extend over a broad frequency range are a key spectral signature of protonated water clusters. They are observed for many membrane proteins that contain internal water molecules, but their microscopic mechanism has remained unclear. Here we compute infrared spectra for protonated and unprotonated water chains, discs, and droplets from ab initio molecular dynamics simulations. The continuum bands of the protonated clusters exhibit significant anisotropy for chains and discs, with increased absorption along the direction of maximal cluster extension. We show that the continuum band arises from the nuclei motion near the excess charge, with a long-ranged amplification due to the electronic polarizability. Our experimental, polarization-resolved light–dark difference spectrum of the light-driven proton pump bacteriorhodopsin exhibits a pronounced dichroic continuum band. Our results suggest that the protonated water cluster responsible for the continuum band of bacteriorhodopsin is oriented perpendicularly to the membrane normal.

[1] Department of Physics, Freie Universität Berlin, 14195 Berlin, Germany. [2] Max-Planck-Institut für Kohlenforschung, 45470 Mülheim an der Ruhr, Germany. [3] Institute of Molecular Science (ICMol), Universitat de València, 46980 Paterna, Spain. Jan O. Daldrop and Mattia Saita contributed equally to this work. Correspondence and requests for materials should be addressed to J.H. (email: jheberle@zedat.fu-berlin.de) or to R.R.N. (email: rnetz@physik.fu-berlin.de)

Water is an excellent conductor for protons since the Grotthuss mechanism allows for transport of protons via the fast motion of a $H_3O^+$ defect[1,2]. The potential of water to efficiently conduct protons is used in various technological applications such as fuel cells and electrochemical devices[3–5]. Proton transfer events are key for many essential biological functions; consequently, one finds protein-bound water molecules and internal water wires in proton-conducting transmembrane proteins such as cytochrome $c$ oxidase, photosystem II, bacteriorhodospin, and channelrhodopsin[6–9].

Because a proton easily moves from one water molecule to a neighboring one, which is a consequence of low barriers in the proton energy landscape[10–13], the infrared (IR) absorption spectrum of an excess proton in water is not characterized by sharp bands but rather by very broad, so-called continuum bands. The first IR continuum band was detected in concentrated acid solutions[14] and later seen in a host of different bulk systems[10].

A well-studied transmembrane protein where protein-bound water molecules play a crucial role is the light-driven proton pump bacteriorhodopsin (bR)[7,15–18]. For bR, light-induced broad IR absorption bands have been observed to rise and decay during the photocycle[19–22]. The interpretation of these broad bands is subtle due to several complications: (i) Three water clusters exist in bR, so it is a priori not clear which water cluster gives rise to which spectral feature[22]. (ii) The translocation of a proton from the cytoplasmic to the extracellular side involves a number of transient states. (iii) More than one excess proton presumably is present at a time. (iv) In the wavenumber range 2500–3000 cm$^{-1}$, broad bands have been assigned to strongly hydrogen-bonded water molecules that do not contain an excess proton[23]. (v) The energy dissipated from the retinal excitation might be absorbed by bulk water molecules, also generating transient broad bands[20].

The continuum band of bR observed in the 1800–2200 cm$^{-1}$ range, devoid of any other spectral contributions, has been interpreted in terms of protons that are delocalized over a hydrogen-bonded network of amino acid side chains and water molecules. Yet, the precise molecular origin of this continuum band is still under debate. One plausible scenario involves a group of water molecules close to the extracellular protein surface where the proton is released[24,25]. Therefore, we will assume from here on that the continuum band is generated by this protonated cluster of water molecules. An alternative scenario involves a proton that is shared between glutamate residues[26], see ref. [22] for a recent overview.

In this paper, we compute the IR spectra for protonated and unprotonated water clusters from ab initio molecular dynamics (AIMD) trajectories on the BLYP/TZV2P level using previously established techniques[27]. We compare the spectra for water clusters that are linear, and essentially consist of a single water chain, with two-dimensional water discs and three-dimensional water droplets. We demonstrate that the continuum band in non-spherical protonated water clusters is anisotropic and that such an anisotropy is experimentally detectable in proton-conducting transmembrane proteins, which allows to draw conclusions on the shape and orientation of the protonated water cluster within the protein. In our experimental IR spectra, we resolve the continuum band of bR aligned in oriented purple membranes. We demonstrate its linear dichroism with a preferred orientation in the plane orthogonal to the membrane normal. This result is consistent with a protonated water cluster extended perpendicularly to the direction of proton pumping. Based on our ab initio simulations, we perform a local spectral analysis of the protonated water chain, which shows that the continuum band arises from polarization fluctuations of the excess proton, which is predominantly observed in a Zundel complex as it moves axially along the chain. The electronic polarizability of the chain gives

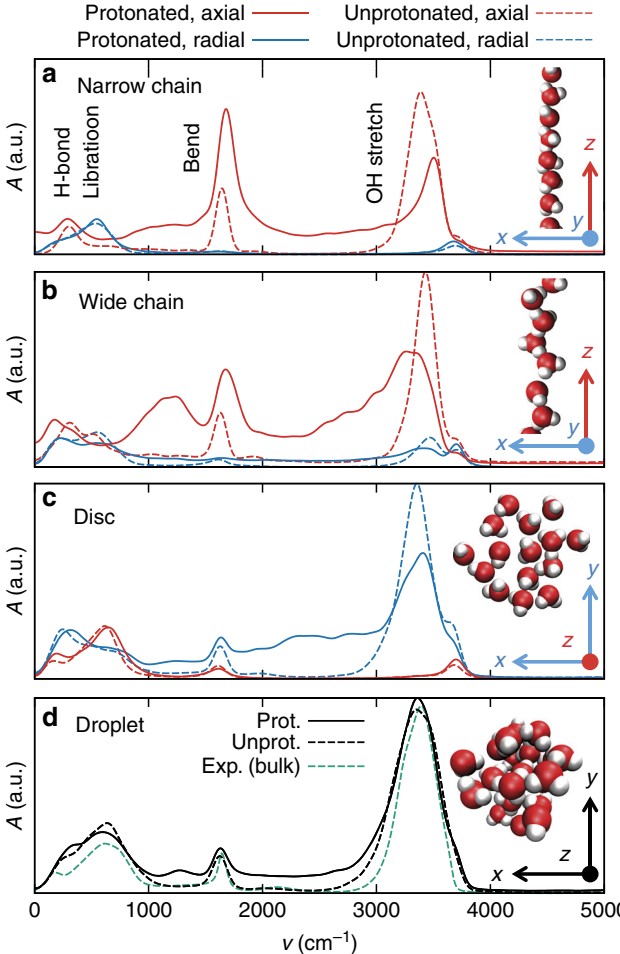

**Fig. 1** Simulated spectra and snapshots. Anisotropic infrared spectra computed from ab initio trajectories for **a** a narrow chain, **b** a wide chain, **c** a disc, and **d** a droplet of water with (solid lines) and without (dashed lines) an excess proton. Red lines show the spectrum for an $E$ field in the axial $z$ direction, blue lines the spectrum for an $E$ field in the radial $xy$ directions, where radial components are averaged. For the droplet, all directions are averaged. In **d**, the experimental bulk water absorption spectrum at $T = 25\,°$C[31] is included (green dashed line). In each panel, the intensities of the computed spectra are drawn to scale

rise to a moderate but long-ranged amplification of the polarization fluctuations.

## Results

**Anisotropic spectra from ab initio simulations**. In the ab initio simulations, we consider three different water cluster geometries, namely, chains, discs, and droplets, consisting of 15, 15, and 26 water molecules, respectively. The different water cluster geometries are stabilized by harmonic confining potentials with strengths such as to produce realistic water densities (see Methods section). For the chain geometry, we compare a narrow and a wide version; configurational snapshots are shown in Fig. 1. Similar simulations have been previously performed for water chains and fast proton transport has been observed and characterized[28,29]. The computed IR spectra are shown in Fig. 1, where we compare spectra in the presence of an excess proton (solid lines) to results without an excess proton (dashed lines). For chains and discs, we distinguish between absorption with the $E$ field parallel to $z$, which is the rotational symmetry axis (red lines), and absorption with the $E$ field in the $xy$ plane (blue lines).

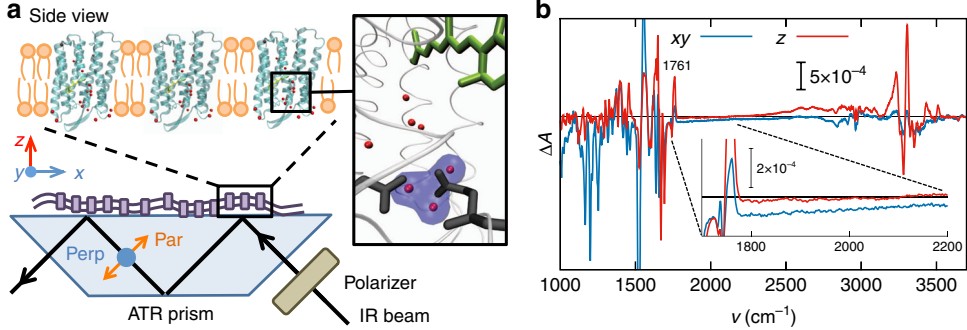

**Fig. 2** Experimental polarization-resolved spectra. **a** In the experimental set-up, the purple membranes are oriented in the $xy$ plane and the proton pumping direction is along $z$. The black arrow indicates the direction of the probing IR light, which is polarized parallel or perpendicular with respect to the plane of incidence. The inset shows a zoom-in of the crystal structure of bacteriorhodopsin[70], with three water molecules highlighted in blue, which presumably cause the continuum band[22, 25]. **b** Experimental IR light-minus-dark difference spectra calculated along the $xy$ and $z$ directions over the whole measured range between 1000 and 3800 cm$^{-1}$. In the inset, the enlarged difference absorption spectra in the relevant wavenumber range 1700–2200 cm$^{-1}$ are shown together with the (positive) band at 1761 cm$^{-1}$ of the C=O stretching vibration of D85

Comparison between the absorption along equivalent directions (i.e., $x$ and $y$ directions for chains and discs) allows us to estimate statistical errors (see Supplementary Figs. 1 and 2).

In all geometries, we resolve well-separated bands that correspond to the OH stretch vibration around wavenumbers $\nu \approx 3400$ cm$^{-1}$, the H$_2$O bending mode around $\nu \approx 1645$ cm$^{-1}$, and bands at much lower wavenumbers that are due to hydrogen bond vibrations and librations[30]. We compare the simulated water droplet spectra in Fig. 1d (black lines) with the experimental absorption spectrum of liquid water[31] (green dashed line). Although our ab initio simulations neglect quantum nuclei motion, which in principle can be corrected for heuristically[32], the numerical spectrum for the unprotonated water droplet (black dashed line) reproduces the experimental band positions and shapes quite nicely.

For all geometries, the presence of an excess proton gives rise to a broad and pronounced continuum band extending over almost the entire wavenumber range (protonated minus unprotonated difference spectra are shown in Supplementary Fig. 3). Most importantly, the continuum band is almost completely polarized along the direction of maximal cluster extension, i.e., along the $z$ axis for chains in Fig. 1a, b and along the $xy$ plane for the disc in Fig. 1c. This resonates well with the intuitive notion that the continuum band is caused by the motion of a delocalized proton, which due to the confinement occurs along the $z$ axis for a water chain and in the $xy$ plane for the disc. Although the anisotropy of the continuum band is more pronounced for the narrow chain in Fig. 1a, it is still significant for the wide chain in Fig. 1b, which demonstrates that the continuum band anisotropy is robust with respect to the water chain structure.

A closer look at the spectra reveals that the high wavenumber shoulder of the OH stretch vibration peak, which is ascribed to dangling OH bonds[33,34], comes from polarization fluctuations that are perpendicular to the cluster extension, both in the presence and absence of an excess proton. This nicely confirms the intuitive expectation, since dangling OH bonds are predominantly oriented radially for chains and axially for discs. For the bending mode, the opposite anisotropy is observed, demonstrating that the water dipole moment (and thus the predominant absorption in the bending mode) points along the direction of maximal extension for the non-spherical clusters. This shows that polarization resolved IR spectroscopy of non-spherical aligned water clusters allows to obtain detailed information on the water cluster structure and orientation.

We conclude that an excess proton gives rise to a pronounced continuum band, regardless of the water cluster geometry. This in turn means that the mere presence of a continuum band in an IR spectrum conveys little information on the proton-solvating water cluster geometry. However, and as we will show now by comparison with our experimental results for bR, the pronounced polarization anisotropy of the continuum band can be used to reveal whether the protonated water cluster is non-spherical and if so what its orientation is.

**Polarization-resolved experimental IR spectra**. We performed Fourier-transform infrared (FTIR) difference spectroscopic experiments on purple membrane films using the attenuated total reflection (ATR) method[35]. The purple membranes spontaneously form parallel stacks of membrane sheets with the membrane normal along the $z$ direction upon drying on the ATR silicon surface[36], as schematically shown in Fig. 2a. The sample was kept in a controlled semi-dry state (see Methods section), which reduces the amount of bulk water molecules[37] and slows down the bR photocycle[38] but preserves the internal water molecules. As a control, a more hydrated sample was prepared, which resulted in almost indistinguishable IR difference spectra, although of smaller intensity (see Supplementary Figs. 4 and 5). Light absorption by the chromophore triggers a sequence of transitions that have been well characterized by experimental and theoretical means[39]. According to the generally accepted view, in the dark state, an excess proton resides in a water cluster close to the exit site at the extracellular side of the protein, as schematically indicated in the inset of Fig. 2a. Continuous illumination leads to the accumulation of the intermediate M state under our experimental conditions (with only minor contributions from the subsequent N state)[40,41], where the excess proton is not present in the proton release group but has just been transferred to the extracellular medium. Thus the light-induced difference spectrum of bR under continuous illumination contains differential information about the proton release group.

We employed IR light with polarization parallel and perpendicular to the plane of incidence[42]; see Fig. 2a for a schematic illustration of the geometry. The light-minus-dark difference spectrum measured with perpendicularly polarized IR radiation provides the difference spectrum in the $xy$ plane of bR, denoted as $\Delta A_{xy}$, while parallel IR light gives rise to a linear combination of the difference spectra in the $xy$ plane and along the $z$ direction (the latter being denoted as $\Delta A_z$)[43]. The difference spectrum $\Delta A_z$ is calculated according to a published formalism[42,44]. Both spectra $\Delta A_{xy}$ and $\Delta A_z$ are presented in Fig. 2b. The continuum band in the frequency range 1700–2200 cm$^{-1}$ is enlarged in the

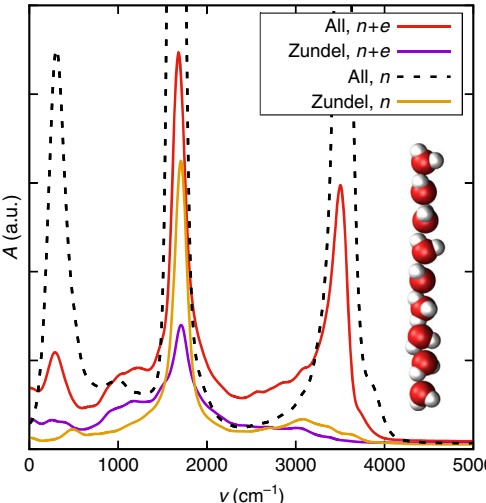

**Fig. 3** Spectral decomposition for a protonated chain. Comparison of the simulated infrared spectrum of the protonated narrow chain in the axial direction including electronic degrees of freedom ($n + e$, red line) to the spectrum obtained from nuclei only by assigning effective charges $q_H = -q_O/2 = 1e$ ($n$, broken line), the spectrum of the projected Zundel ensemble including electronic degrees of freedom (violet line), and the spectrum calculated from the Zundel ensemble using nuclei only (yellow line)

inset. We restrict our analysis to this range, because it is of diagnostic value for the presence of a protonated hydrogen-bonded network of water molecules. In contrast, the dichroism of the broad band >2500 cm$^{-1}$ presumably arises from a neutral cluster of strongly hydrogen-bonded water molecules located at the cytoplasmic side of the protein[21]. Note that the sharp band at 1761 cm$^{-1}$ at the lower end of the continuum band arises from the C=O stretching vibration of the aspartic acid D85, which is the proton acceptor of the retinal Schiff base[45].

The continuum band is clearly evident in the lateral difference spectrum $\Delta A_{xy}$ with a near-constant negative intensity but almost completely absent in the normal difference spectrum $\Delta A_z$ (Fig. 2b, inset). As expected, the continuum band in the light-minus-dark spectrum $\Delta A_{xy}$ is negative, since the corresponding water cluster is protonated in the dark state and deprotonated in the M intermediate state (which dominates under illumination). The recorded FTIR spectra are thus not consistent with a protonated water wire along the $z$ direction, in which case $\Delta A_z$ should be negative, nor with a protonated isotropic water cluster, in which case both $\Delta A_z$ and $\Delta A_{xy}$ should be negative. Rather, this experimental result suggests a delocalized proton in the dark state of bR that is delocalized in the $xy$ plane perpendicular to the pumping direction. This delocalization could in principle originate from a water wire perpendicular to the $z$ axis or from a disc oriented in the $xy$ plane. We also cannot exclude a contribution from a delocalized proton that moves laterally at the surface of the protein[46] or at the surface of the purple membrane, although amino acid exchange studies of the protein[47] invalidate this hypothesis. It is known that protons exhibit enhanced residence times at the surface of lipid membranes[46,48,49], but we expect the surface concentrations of protons in the light and dark state to be rather similar due to the excess of buffer molecules present as well as due to the buffering effect of the protein surface itself. Therefore, we suggest that the contribution of delocalized protons on the membrane surface to the difference spectrum is negligible.

**Microscopic characterization of the continuum band**. To learn more about the origin of the continuum band in protonated water

clusters, we compare in Fig. 3 the simulated axial spectrum of a narrow protonated water chain including contributions from vibrations of the nuclei and electron polarization fluctuations (denoted as $n + e$ and already shown in Fig. 1a, red line) with the spectrum obtained from the nuclear motion alone (denoted by $n$, broken line); for the latter, we assign effective charges $q_H = +e$ and $q_O = -2e$ to protons and oxygens. This choice overestimates the water dipole moment and thus exaggerates the water vibration bands but gives the excess proton the correct charge (see Supplementary Note 1 for details). All spectra are plotted to scale and thus can be compared with each other. Figure 3 shows that in the wavenumber range $\nu \approx 2000$–$3000$ cm$^{-1}$, the amplitude of the continuum band in the full $n + e$ spectrum is doubled compared to the $n$ spectrum, indicating a sizable electronic contribution to the continuum band. We also compute the spectral contribution of the excess proton together with its two neighboring water molecules, which together make up the Zundel complex $O_2H_5^+$ (see Supplementary Note 2 for details on the projection formalism[27,50]). Note that the Zundel complex is the dominant solvation state of the excess proton in water chains[28], in contrast to discs and bulk water, where the Eigen state dominates (see Supplementary Note 3 for a comparison of Zundel occupation probabilities in the different water cluster geometries). In Fig. 3, the contribution of the Zundel complex to the axial spectrum is shown including contributions from nuclear motion and electronic polarization (denoted $n + e$, violet line) and nuclear motion alone (denoted $n$, yellow line). Interestingly, the nuclear motion spectrum of the Zundel complex (yellow line) is sufficient to describe the nuclear motion spectrum of the entire chain (dashed black line) in the range between $\nu \approx 2000$–$3000$ cm$^{-1}$. At the same time, the difference between the Zundel spectrum including only nuclear motion (yellow line) and the Zundel spectrum including also the electronic contribution (violet line) is minimal in this frequency range. This means that the nuclear contribution to the continuum band comes from a very localized region around the excess proton, while the electrons in the Zundel complex contribute little to the spectrum. We conclude that the axial continuum band in protonated water chains is caused by local nuclear polarization fluctuations in the immediate vicinity of the excess proton, which predominantly corresponds to a Zundel complex. The nuclear polarization fluctuations are amplified by the rather long-ranged electronic polarizability, which extends further out to next-nearest neighboring water molecules.

## Discussion

We have computed the IR spectra for protonated and unprotonated water clusters with different geometries from AIMD trajectories, namely, water clusters that are linear and essentially consist of a single water chain with two-dimensional water discs and three-dimensional water droplets. The IR spectra of all protonated water clusters, but not the neutral ones, exhibit pronounced and broad continuum bands, irrespective of the shape. Therefore, the mere existence of a continuum band contains no information about the shape of the protonated water cluster. However, the IR continuum band exhibits strong dichroism for chains and discs, i.e., the absorption is maximal for IR light that is polarized axially for the water chain and in a radial direction for the water disc. For spherical water droplets, the continuum band is isotropic, as expected. This means that the continuum band appears for IR radiation that is polarized along the direction of proton motion.

This anisotropy of the continuum band is not only interesting per se but also aids in the interpretation of the continuum band of bR: different water clusters that exist in bR and which play a role at different stages of the proton transfer reaction presumably have

different shapes and orientations. Therefore, measurements of the IR absorption dichroism allow to exclude certain proton transfer scenarios. In fact, our experimental polarization-resolved IR absorption difference spectra of oriented bR molecules in native purple membranes show a pronounced continuum band with a polarization perpendicular to the membrane normal, suggesting that the protonated water cluster responsible for the signal is predominantly oriented perpendicularly to the membrane normal. To be more specific, while at the cytoplasmic side of bR, where protons enter, water forms a linear water chain that is parallel to the proton transfer direction, the water cluster close to the extracellular release site was indeed speculated to be more likely oriented perpendicularly to the proton transfer direction, i.e., parallel to the membrane[22].

The continuum band has previously been intensely studied by time-resolved FTIR spectroscopy using unpolarized[16,19,20,41,47,51] and, to less extent, polarized[52] IR radiation. Here we resolve its polarization dependence under equilibrium continuous illumination conditions, which excludes spectral artifacts by laser-induced heat transfer from the protein to the aqueous solution[20].

Our decomposition of the numerically calculated spectra into the separate contributions stemming from nuclear dynamics (i.e., actual vibrations) and from electronic polarization effects shows that the continuum band is mostly caused by nuclear motion, which is amplified by slaved electronic polarization effects. A further projection onto the immediate vicinity of the excess proton suggests that indeed the continuum band is caused by the moving excess proton, in agreement with previous results[10,50].

We note that our simulations neglect the coupling to flanking amino acid side chains, which has been demonstrated to be important[26], and instead represent the chemical confinement by a smooth and structureless external potential. More detailed simulations are planned, but we do not expect the basic qualitative features concerning the anisotropy of the continuum band to be dependent on the detailed description of the chemical environment.

Our experimental approach is general and can be used to study the orientation of protonated water clusters also in other membrane proteins, particularly those that involve proton translocation, like, e.g., photosystem II[53], photosynthetic reaction center[54,55], archaerhodopsin-3[56], and cytochrome $c$ oxidase[57]. Apart from proteins, we expect similar effects also in other systems that contain water clusters, for example, in inverted hexagonal lipid phases, where water forms hexagonally ordered parallel cylinders, or in lipid lamellar phases, where water forms thin slabs[58]. Such lipid systems are known to orient on surfaces[59], which enables IR dichroism studies with similar techniques as used by us. A different system that contains thin water chains are subnanometer diameter carbon nanotubes that can be embedded and thereby oriented in lipid membranes[60].

## Methods

**Ab initio simulations.** The AIMD simulations are carried out with the CP2K 2.5 software package[61], using a TZV2P basis set[62] and the BLYP exchange correlation functional[63]. Box sizes are determined from the extension of typical snapshots of the systems so that each atom is at least 0.8 nm away from the nearest boundary. For all systems, we use non-periodic boundary conditions based on the Martyna–Tuckerman method[64]. For the production runs, we first generate 15 ps trajectories using a massive Nosé–Hoover chain thermostat[65,66] (chain length of 3) to generate an equilibrium ensemble at $T = 300$ K. From multiple snapshots of this trajectory, we start NVE runs of about 5 ps each to obtain a total NVE simulation time of 130 ps per system.

**Calculation of IR spectra.** For the calculation of the molecular dipole moments, localized Wannier centers for all electron pairs are computed and saved in each

simulation step. IR spectra are computed from the trajectories via

$$A(\omega) \propto \int \langle \dot{\mu}(0)\dot{\mu}(t) \rangle e^{-i\omega t} \, dt, \qquad (1)$$

where $\mu$ denotes the dipole moment computed from the nuclei and Wannier center positions. For visualization, the spectra are smoothed by a Gaussian kernel with a width of $\sigma_\nu = 50$ cm$^{-1}$.

**Simulation system setup.** The oxygen atoms of the water chains are radially constrained by a harmonic potential $U_{xy}^{\text{chain}}(x, y) = K_{xy}(x^2 + y^2)/2$, with $K_{xy} = 2000$ kJ mol$^{-1}$ nm$^{-2}$ for the narrow chain and $K_{xy} = 20$ kJ mol$^{-1}$ nm$^{-2}$ for the wide chain. Note that the oriented water molecules in the chain weakly localize the excess proton in the chain center[67].

The disc system is axially constrained by $U_z^{\text{disc}}(z) = K_z z^2/2$, with $K_z = 2000$ kJ mol$^{-1}$ nm$^{-2}$ and laterally by a soft potential of the form $U_{xy}^{\text{disc}}(x, y) = K_{xy}(x^2 + y^2)/2$ with $K_{xy} = 30$ kJ mol$^{-1}$ nm$^{-2}$. The number of water molecules and the force constant $K_{xy}$ are determined from the radial distribution function of a periodic two-dimensional water slab simulated using classical force fields, as explained in Supplementary Note 4.

The droplet system is constrained by a soft isotropic potential $U^{\text{drop}}(x, y, z) = K_r(x^2 + y^2 + z^2)/2$ with $K_r = 40$ kJ mol$^{-1}$ nm$^{-2}$, see Supplementary Note 4.

**Experimental sample preparation.** Purple membrane patches containing the light-driven proton pump bR were prepared from strain S9 of *Halobacterium salinarum*[68]. Approximately 8 μl of protein solution (5 mg per ml dissolved in an aqueous solution of 3 mM NaCl, 3 mM Hepes buffer at pH 7.0) was dried on the ATR crystal under a gentle dry air stream. The native membrane patches orient preferentially along the Si surface[36]. The IR absorption of the dry sample was around 0.8 in the amide I region at 1657 cm$^{-1}$. The protein film was rehydrated through the saturated water vapor phase (see ref. [41] for experimental details) of a glycerol/water mixture (8:2 wt/wt) resulting in 46% relative humidity according to ref. [69]. The bR sample contained about 750 molecules of water per bR monomer. The comparison to a more hydrated sample is described in Supplementary Methods. The protein absorption spectrum is shown in Supplementary Fig. 6.

**Polarized ATR/FTIR spectroscopy.** An ATR accessory using silicon as material for the internal reflection element (Smiths detections) was placed into the FTIR spectrometer (Vertex 80v, Bruker). Light–dark difference spectra were recorded during illumination (light-emitting diode emitting at 530 nm) at a spectral resolution of 2 cm$^{-1}$. The polarized probe beam was generated by a grid polarizer (Holographic wire grid, Thorlabs) at two perpendicular orientations. The light-induced difference spectra recorded with parallel and perpendicularly polarized light are shown in Supplementary Fig. 4. The $xy$ and $z$ light–dark difference spectra were calculated according to refs. [42–44] with an angle of incidence of 39° of the IR beam in the ATR crystal surface and an index of refraction of the rehydrated membrane protein film of $n = 1.6$. The angle of incidence has been calculated[43] from the measured dichroic ratio of $R_{\text{ATRiso}} = 2.1$ for water (data not shown). We have calculated the $xy$ and $z$ spectra assuming different incident angles ($\alpha = 39°$, 41°, 37°) and sample refraction indices ($n = 1.7$ or 1.5), showing that our uncertainties in these values do not have a significant impact on the calculated $xy$ and $z$ spectra (see Supplementary Fig. 7).

**Data availability.** The datasets generated and analyzed during the current study are available from the corresponding authors on reasonable request.

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

## Acknowledgements

We gratefully acknowledge financial support from the DFG (Grant No. SFB 1078, projects B3 and C1). M.H. is grateful for support from the Cluster of Excellence RESOLV (EXC 1069) funded by the DFG. V.A.L.-F. thanks MINECO for the financial support (Grant No. BFU2016-768050-P and Fellowship No. RYC-2013-13114).

## Author contributions

J.O.D. and M.S. contributed equally to this work. J.O.D., R.R.N., and J.H. designed the study. J.O.D. and M.H. set up the simulations. J.O.D. carried out the simulations and data analysis. M.S., V.A.L.-F., and J.H. designed the experimental set-up. M.S. performed the experiments. J.O.D., M.S., and R.R.N. wrote the manuscript with contributions from all authors.
