## [Peer Review File · Nature Communications]

Reviewer #1 (Remarks to the Author):

The mechanism of proton transport is a central problem of bioenergetics, the details of proton motion have been subject of theoretical and experimental studies for a long time. The present work introduces a new approach in the field: polarized infrared vibrational spectroscopy, with experimental and theoretical approach. The object is the light driven transmembrane proton pump bacteriorhodopsin. This system has been extensively studied with respect to the molecular events during proton transport. It has been shown that both protonated protein side chains and water molecules play key role in the proton transport. Water clusters inside the protein with specific structure have been suggested in various stages of the pumping cycle. Vibrational IR spectroscopy has proven very fruitful in these studies.

The present paper brings a new twist to the IR spectroscopy based investigations: polarized vibrational spectroscopy is used to assign specific protonated water cluster structures inside the protein. It is shown by ab initio calculations that protonated water clusters with anisotropic shape will have anisotropic IR absorption, several characteristic structures are analyzed in detail. IR polarized spectroscopy was performed on highly oriented samples of purple membranes, using ATR technique. High anisotropy was found in difference spectra characteristic of the bR to M state transition, and comparing theory and experiments likely water cluster formations were identified. The paper is based on a good idea. It is carried out properly, both experimentally and theoretically, the conclusions are sound. It is a clever and solid work. There is a question, however, about the possible generalization of the method – a necessity for publication in the high profile interdisciplinary journal. It is hard to imagine that the method will be used on a different system soon. Bacteriorhodopsin is a very unique system that makes such experiments possible: it can be partially dried so that buried water can be well seen, the material is extremely sturdy, the 2D crystalline purple membrane structure makes the preparation of highly oriented samples possible, all properties that are necessary for these experiments to succeed, and they are not present on practically any other protein. Still, the paper represents real novelty, the results are good, so it can be published in Nature Communication.

Reviewer #2 (Remarks to the Author):

The authors report the fascinating finding that protonated water wires/clusters have highly anisotropic infrared absorption. This feature can potentially be used to help figure out the structure and possibly composition of (protonated) water clusters in proteins; this is of great mechanistic relevance to proteins that conduct ions/protons. In fact, the concept was demonstrated by an experimental study of bacteriorhodopsin in oriented membrane. The data suggest that the protonated cluster in the "proton release group" is likely oriented perpendicular to the membrane normal, providing additional clues to the structural feature of this important structural motif. Finally, computational analysis suggested that the continuum band is mostly dictated by the proton delocalization; while electronic polarization contributes, the effect is relatively minor.

I think the contribution is of great fundamental nature and should be of interest to the readers of Nature Communication. The ms is generally clearly written.

I have one question regarding the experimental data in Fig. 2b. There is a broad positive feature in the z-component absorption near 2500 cm^{-1} , and it is absent from the xy component. What is the origin of this feature?

Please see our responses below, along with reproduced reviewers' comments in *blue italics*. All changes in the manuscript that address referees' comments are highlighted in red.

Reviewer #1 (Remarks to the Author):

The mechanism of proton transport is a central problem of bioenergetics, the details of proton motion have been subject of theoretical and experimental studies for a long time. The present work introduces a new approach in the field: polarized infrared vibrational spectroscopy, with experimental and theoretical approach. The object is the light driven transmembrane proton pump bacteriorhodopsin. This system has been extensively studied with respect to the molecular events during proton transport. It has been shown that both protonated protein side chains and water molecules play key role in the proton transport. Water clusters inside the protein with specific structure have been suggested in various stages of the pumping cycle. Vibrational IR spectroscopy has proven very fruitful in these studies.

The present paper brings a new twist to the IR spectroscopy based investigations: polarized vibrational spectroscopy is used to assign specific protonated water cluster structures inside the protein. It is shown by ab initio calculations that protonated water clusters with anisotropic shape will have anisotropic IR absorption, several characteristic structures are analyzed in detail. IR polarized spectroscopy was performed on highly oriented samples of purple membranes, using ATR technique. High anisotropy was found in difference spectra characteristic of the bR to M state transition, and comparing theory and experiments likely water cluster formations were identified.

The paper is based on a good idea. It is carried out properly, both experimentally and theoretically, the conclusions are sound. It is a clever and solid work. There is a question, however, about the possible generalization of the method – a necessity for publication in the high profile interdisciplinary journal. It is hard to imagine that the method will be used on a different system soon. Bacteriorhodopsin is a very unique system that makes such experiments possible: it can be partially dried so that buried water can be well seen, the material is extremely sturdy, the 2D crystalline purple membrane structure makes the preparation of highly oriented samples possible, all properties that are necessary for these experiments to succeed, and they are not present on practically any other protein. Still, the paper represents real novelty, the results are good, so it can be published in Nature Communication.

We agree with the reviewer that bacteriorhodopsin in purple membranes is unique in terms of orientation and stability. Nevertheless, we envision to apply the methods introduced in our manuscript to other membrane proteins as well. In the revised manuscript on page 5, we added a comment including a referenced list of other membrane proteins where broad bands have been assigned to delocalized excess protons. In that paragraph we also mention that our methods may be applied to protonated and unprotonated water chains and water discs in lamellar systems as well as to water chains in aligned sub-nm diameter carbon nanotubes, which have been recently embedded into lipid membranes (Tunuguntla et al. (2016), Nat. Nanotechnol. 11, 639).

Reviewer #2 (Remarks to the Author):

The authors report the fascinating finding that protonated water wires/clusters have highly anisotropic infrared absorption. This feature can potentially be used to help figure out the structure and possibly composition of (protonated) water clusters in proteins; this is of great mechanistic relevance to proteins that conduct ions/protons. In fact, the concept was demonstrated by an experimental study of bacteriorhodopsin in oriented membrane. The data suggest that the protonated cluster in the "proton release group" is likely oriented perpendicular to the membrane normal, providing additional clues to the structural feature of this important structural motif. Finally, computational analysis suggested that the continuum band is mostly dictated by the proton delocalization; while electronic polarization contributes, the effect is relatively minor.

I think the contribution is of great fundamental nature and should be of interest to the readers of Nature Communication. The ms is generally clearly written.

I have one question regarding the experimental data in Fig. 2b. There is a broad positive feature in the z-component absorption near 2500 cm⁻¹, and it is absent from the xy component. What is the origin of this feature?

We thank the reviewer for pointing out that the experimental feature around 2500 cm⁻¹ was not clearly explained in the submitted manuscript. Bacteriorhodopsin has, besides a protonated water cluster (the proton release group), at least two neutral water clusters that are involved in proton translocation, as was mentioned in the introduction of the manuscript. One of these clusters is transiently formed during the photocycle of bR. It has been proposed that three water molecules transiently form a chain between Asp96 and the Schiff base (i.e., normal to the membrane surface), allowing for an H-bonded network connecting both groups through which proton transfer would occur (Freier et al. (2011), PNAS 108, 11435–11439). Experiments and simulations indicate that the transient formation of this chain leads to a broad positive band around 2500 cm⁻¹, stemming from H-bonded OH vibrations of these water molecules (Freier et al. (2011), PNAS 108, 11435–11439, Wolf et al. (2014), J. Chem. Phys. 141, 22D524). This water cluster does not store a proton, but it is supposed to transfer a proton only transiently via the Grotthuss mechanism. The transient protonation takes place on the ps timescale, and is thus much too short to contribute to our experimental time-averaged difference spectra shown in Fig. 2b. Our experimental observation that the band around 2500 cm⁻¹ is predominantly polarized along the z direction (see Fig. 2b) is consistent with the assignment of this band to a water chain normal to the membrane surface. In fact, this interpretation also agrees with our simulations, since the calculated spectra of a neutral wide water chain (dashed lines in Fig. 1b) show a weakly increased IR absorption intensity along the z axis (compared to the xy-directions) in the entire IR frequency range. This discussion further illustrates the high potential of polarization experiments combined with anisotropic spectral simulations to characterize water clusters in proteins, whether protonated or neutral.

The present manuscript focuses on the proof-of-concept that polarization-resolved IR spectroscopy in a combined simulation-experimental approach is a useful tool to study water clusters in proteins. We feel that an in-depth discussion of additional features of the bacteriorhodopsin photocycle related to unprotonated water clusters might be more confusing than helpful. We added a sentence on page 4 that explains why we focus our discussion on the spectral interval 1800 - 2000 cm^{-1} .

Additional changes

We decided that a shared first authorship between J.O. Daldrop (who did the theoretical work) and M. Saita (who carried out the experiments) is more appropriate. Additionally, we incorporated a few minor editorial changes, added the Ref. [51] (Garczarek et al. (2005), PNAS 102, 3633–3638) to the revised manuscript, and changed the color coding in Fig. S10 in the SI.

We hope that the revised manuscript addresses the concerns raised by the reviewers and that it is now suitable for publication in Nature Communications.

Reviewer #1 (Remarks to the Author):

With the additions the paper can now be accepted for publication.

Reviewer #2 (Remarks to the Author):

The authors made revisions to adequately respond to the previous round of review.